# New Carboxylate Anionic Sm-MOF: Synthesis, Structure and Effect of the Isomorphic Substitution of Sm³⁺ with Gd³⁺ and Tb³⁺ Ions on the Luminescent Properties

Anna A. Ivanova [1], Victoria E. Gontcharenko [1,2], Alexey M. Lunev [1], Anastasia V. Sidoruk [1], Ilya A. Arkhipov [1], Ilya V. Taydakov [2] and Yuriy A. Belousov [1,2,*]

[1] Chemistry Department, Moscow State University, Leninskie Gory, 119991 Moscow, Russia; anna.ivanova@chemistry.msu.ru (A.A.I.); victo.goncharenko@gmail.com (V.E.G.); lunev94@yandex.ru (A.M.L.); avs_1999@mail.ru (A.V.S.); ilia.arkhipov@chemistry.msu.ru (I.A.A.)

[2] P. N. Lebedev Physical Institute of Russian Academy of Sciences, 119991 Moscow, Russia; taidakov@gmail.com

\* Correspondence: belousov@inorg.chem.msu.ru

**Abstract:** Two new compounds, namely $\{(NMe_2H_2)\}[Ln(TDA)(HCOO)]$ $0.5H_2O$, Ln = Sm³⁺ (**Sm-TDA**) and Gd³⁺ (**Gd-TDA**), where $TDA^{3-}$ is the anion of 1*H*-1,2,3-triazole-4,5-dicarboxylic acid ($H_3TDA$), were synthesized by the solvothermal method in a DMF:$H_2O$ mixture. According to single-crystal X-ray diffraction data, the compounds are 3d-MOFs with an anionic lattice and dimethylammonium cations occupying part of the cavities. Based on these compounds, two series of mixed-metal complexes, $[NMe_2H_2][Sm_xLn_{1-x}(TDA)(HCOO)]$, (x = 0.9 (**Sm$_{0.9}$Ln$_{0.1}$-TDA**), x = 0.8 (**Sm$_{0.8}$-Ln$_{0.2}$-TDA**) . . . **Sm$_{0.02}$Ln$_{0.98}$-TDA**, Ln = Tb, Gd), were also obtained and characterized by powder XRD. The luminescent properties of the compounds were studied and it was shown that the resulting compounds are two- or three-component emitters with the possibility of fine color tuning by changing the intensities of fluorescence and phosphorescence of the ligand, as well as the luminescence of Sm³⁺ and Tb³⁺ f-ions.

**Keywords:** [1*H*]-1,2,3-triazole dicarboxylic acid; gadolinium; samarium; terbium; luminescence; crystal structure; MOF

## 1. Introduction

Coordination compounds of lanthanides have been proposed as functional materials for luminescent light emitters [1–5], biomedical imaging [6–8], sensors [9–12], luminescent thermometers [13–16], and as materials for the protection of banknotes and security [17] due to their unique luminescent properties corresponding to partially filled f-shells of lanthanide ions. The main strategies to improve the luminescent characteristics of Ln³⁺ complexes are by designing a ligand with suitable antenna characteristics [18,19], reducing the efficiency of vibrational relaxation by minimizing the number of C-H, N-H, and O-H bonds [20], and suppressing concentration quenching by diluting the luminescent Ln³⁺ ion with optically inactive La³⁺ [21,22], Gd³⁺ [20,23], or Y³⁺ ions [5,24,25]. A great deal of work on luminescent lanthanide compounds is devoted to molecular complexes, especially β-diketonates [1,26,27] and acylpyrazolonates [20,28–30]. The transition to metal–organic frameworks (MOF) allows us to significantly expand the possibilities of application in the field of creating sensor materials, but this requires the use of polydentate ligands, such as polycarboxylic acids [31,32]. The advantage of complexes based on aromatic carboxylic acids is the presence of a strong chromophore in the ligand molecule that effectively absorbs the excitation energy, which makes the increase in the luminescence intensity of the complexes [17,31] possible. Due to this, MOFs based on aromatic [33,34] and heteroaromatic carboxylic acids [32,35] have good light absorbing properties, as well as thermal stability

exceeding that of β-diketonates. To create effectively luminescent Ln-MOF, it is necessary to minimize the number of OH, CH, and NH bonds in the structure of the complex, since they contribute to the effective quenching of the luminescence of $Ln^{3+}$ ions [36]. Some azolpoly-carboxylic acids have these requirements—for example, [1*H*]-1,2,3-triazole-4,5-dicarboxylic acid ($H_3TDA$) [10,23,37–41], for which it is possible to implement various coordination modes [32]. In addition, the fully deprotonated ion $TDA^{3-}$ does not contain CH-, NH-, and OH-groups that cause luminescence quenching. Despite the attractiveness and synthetic availability, lanthanide complexes with $H_3TDA$ have not been studied extensively. Thus, depending on the synthesis conditions, linear polymers {[$Ln(H_2O)_4(HTDA)(H_2TDA)$]} [37–39] containing partially deprotonated ligand anions, and three-dimensional polymers {[$Ln(TDA)(H_2O)_n$]} [10,38,41] with the anion $TDA^{3-}$, can be obtained. Both of these families of compounds contain intraspheric water molecules that reduce the luminescence efficiency. To exclude water from the coordination sphere, we previously proposed synthesis in the presence of DMF, leading to a three-dimensional MOF {($NMe_2H_2$)[$Eu(TDA)(HCOO)$] $0.5H_2O$} [23]. For this compound, the lifetime of the excited state $\tau_{obs}$ of $Eu^{3+}$ ions has been increased by 2.2 times compared to the {[$Eu(TDA)(H_2O)_3$]($H_2O$)} complex containing intraspheric water molecules. Study of the photoluminescent properties of a series of isostructural compounds ($NMe_2H_2$)[$Eu_xGd_{1-x}(TDA)(HCOO)$] $0.5H_2O$ (x = 0.1–0.9) showed that the introduction of $Gd^{3+}$ ions also significantly (by 2.35 times at x = 0.9) increases the lifetime of the excited state of $Eu^{3+}$ ions by suppressing concentration quenching.

$Sm^{3+}$ and $Eu^{3+}$ ions are united not only by the proximity of the ionic radii, but also by a certain similarity in luminescent properties (red-orange color of luminescence, proximity of the energies of resonant levels) [17]. At the same time, the values of the quantum yields of luminescence and the lifetime of the excited state $\tau_{obs}$ of $Sm^{3+}$ complexes in most cases are significantly inferior to those of $Eu^{3+}$ compounds [17,32]. The aim of this work was to synthesize the samarium complex with $H_3TDA$ in the $H_2O$–DMF medium, as well as to establish the possibility of controlling the luminescent properties by partial substitution of $Sm^{3+}$ ions with $Gd^{3+}$ ions and $Tb^{3+}$. As a result of the work, optimal conditions for obtaining pure phases {($NMe_2H_2$)[$Ln(TDA)(HCOO)$] $0.5H_2O$} (Ln = Sm(**Sm-TDA**), Gd(**Gd-TDA**), **$Sm_xGd_{1-x}$**, **$Sm_xTb_{1-x}$**) were found, the crystal structure of complexes **Sm-TDA** and **Gd-TDA** was studied, and the luminescent properties of monometallic compounds **Sm-TDA** and **Gd-TDA** were investigated, as well as a series of mixed-metal complexes, **$Sm_xGd_{1-x}$-TDA**, **$Sm_xTb_{1-x}$-TDA** (x = 0.02–0.9).

## 2. Results

### 2.1. Synthesis and Composition

The high coordination numbers of $Ln^{3+}$ ions are achieved in many cases due to the coordination of water molecules. Synthesis performed in the presence of a non-aqueous polar solvent may lead to the exclusion of water from the coordination sphere. N,N-dimethylformamide in solvothermal conditions can act as a source of $NMe_2H_2^+$ [34,42] and $HCOO^-$ [34,42–44] ions included in the structure of some MOFs. In some cases, the unchanged DMF molecule is embedded in the polymer structure, helping the central atom to achieve a high coordination number [35,45].

It was discovered that during solvothermal synthesis in the $H_2O$:DMF system (1:1), a crystalline precipitate of the composition [($CH_3$)$_2NH_2$]$^+$[$Sm(TDA)(HCOO)$]$^- \cdot 0.5H_2O$ is formed from a mixture of $Sm(NO_3)_3$, $H_3TDA$, and NaOH.

To confirm the composition, complex **Sm-TDA** was investigated by the DTA method with mass spectrometric detection of volatile products (Figure 1). In the range of 42–150 °C, water loss occurs (the loss of less than 0.5 water molecules per one formula unit **Sm-TDA 0.5H_2O** is calc. 2.22%, exp. 1.16%). The substance is stable upon further heating, and only at temperatures above 330 °C does its decomposition begin, followed by oxidation to $Sm_2O_3$, $CO_2$, $H_2O$, and $N_2$ (mass loss: calc. 56.9%, exp. 55.6%) Thus, the decomposition of

dimethylamine occurs only simultaneously with the combustion of the complex, which indicates the high stability of the compound under study.

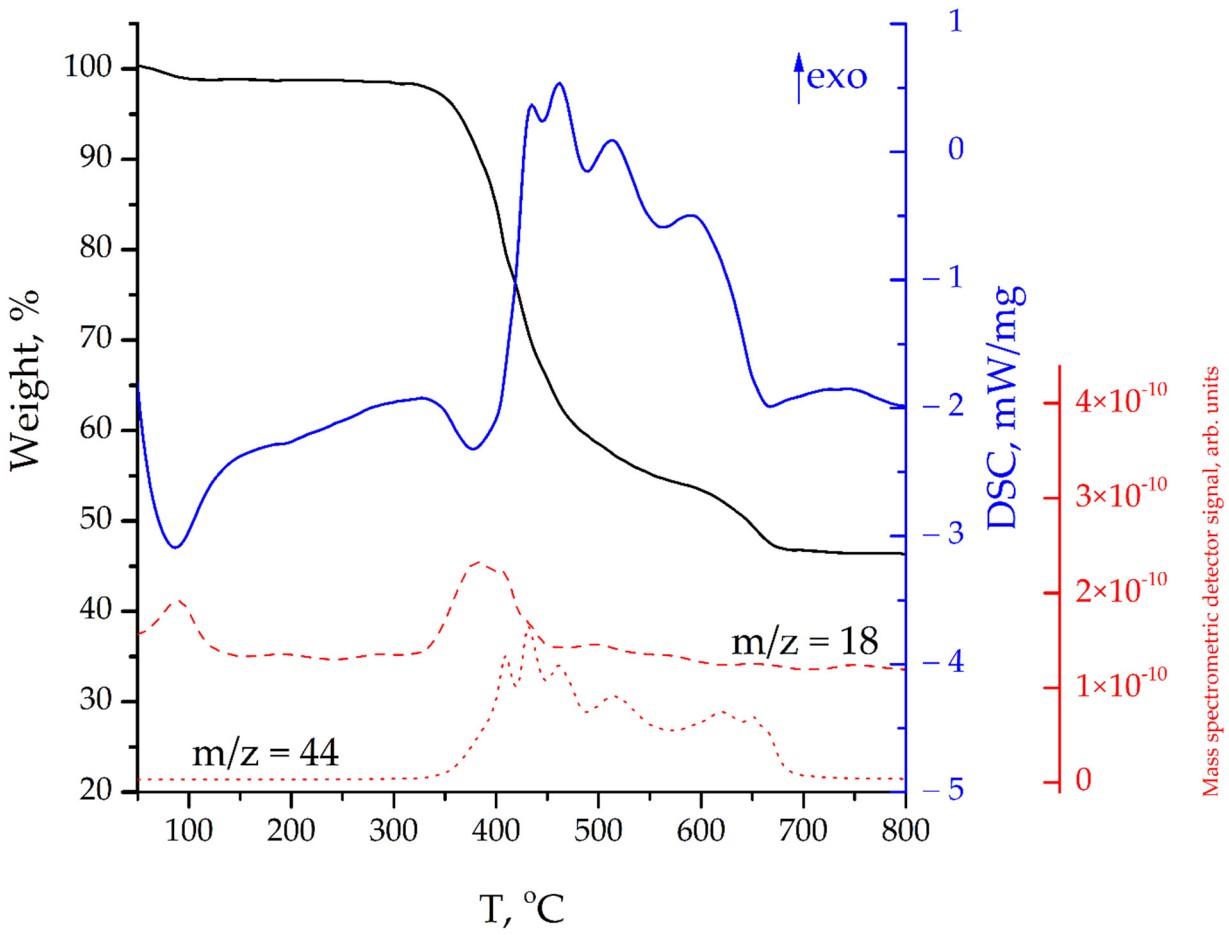

**Figure 1.** Curves of mass loss (TG), DTA, as well as signals from the mass spectrometric detector of thermal decomposition products.

Initially, several attempts to perform single-crystal XRD showed the presence of only crystals of samarium formate isostructural to the described REE formates [46]. PXRD data did not show the presence of this impurity phase in the synthesis product, which can be explained by the presence of a small (<5%) number of large well-formed crystals {Sm(HCOO)$_3$}, which, due to their satisfactory size and shape, were selected for single-crystal XRD. To obtain the pure phase of **Sm-TDA**, a number of syntheses were carried out with varying pH values (1.5–8 in increments of 0.5 $\pm$ 0.1). It was found that synthesis at a pH of 1.5 leads to the formation of large single crystals of **Sm-TDA**, pure from the impurity {Sm(HCOO)$_3$}, not only according to the PXRD, but also optical microscopy (Figure 2). A decrease in the pH of the solution contributes to a decrease in the concentration of the ligand anion TDA$^{3-}$, which slows down the MOF formation reaction and promotes the growth of larger crystals suitable for X-ray. This optimized technique was applied to the synthesis of mixed-metal compounds **Sm$_x$Gd$_{1-x}$TDA** and **Sm$_x$Tb$_{1-x}$TDA**.

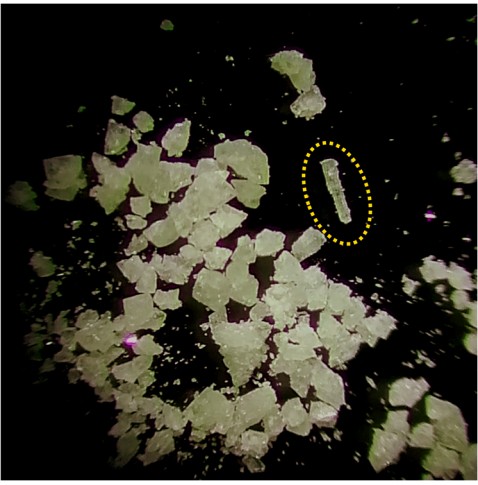 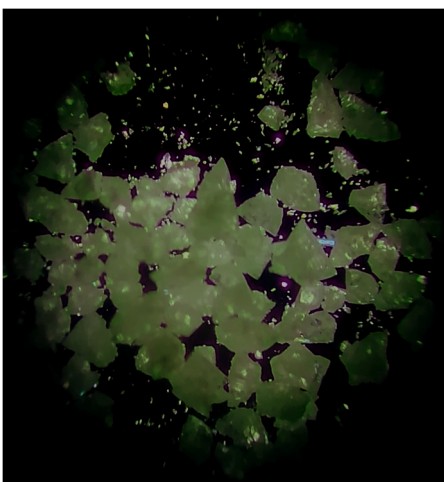

**Figure 2.** Crystals of **Sm-TDA** obtained at pH 5 (**left**) and 1.5 (**right**). The crystal of the impurity {Sm(HCOO)$_3$} is visible in the left photo (marked with a yellow oval).

Due to the proximity of ionic radii, in the absence of thermodynamic control conditions [47], mixtures containing ions of various REE crystallize, as a rule, as mixed-metal compounds with statistical distribution over positions [10,20–25,48,49]. According to the PXRD data, phase separation is not observed in systems **Sm$_x$Gd$_{1-x}$TDA** and **Sm$_x$Tb$_{1-x}$TDA**. According to the EDX of crystals, the ratio of Sm:Gd and Sm:Tb is preserved for individual crystals in the powder. ICP MS spectrometry data confirmed the compliance of the Ln$^1$:Ln$^2$ embedded during synthesis.

All synthesized complexes, **Sm-TDA,** x = 0.1 (**Sm$_{0.9}$Ln$_{0.1}$-TDA**), x = 0.2 (**Sm$_{0.8}$-Ln$_{0.2}$-TDA**) . . . **Sm$_{0.1}$Ln$_{0.9}$-TDA**, Ln = Tb, Gd, were found to be isostructural, which was confirmed by PXRD data. The diffractograms of the compounds correlate well with the theoretical one calculated from the **Sm-TDA** crystal structure data. The PXRD of the series of mixed-metal complexes and the theoretical PXRD calculated for structure **Sm-TDA** are shown in Figure 3. A small difference in the relative intensities of individual reflexes is associated with the texture of the sample and the presence of a variable number of water molecules in the cavities.

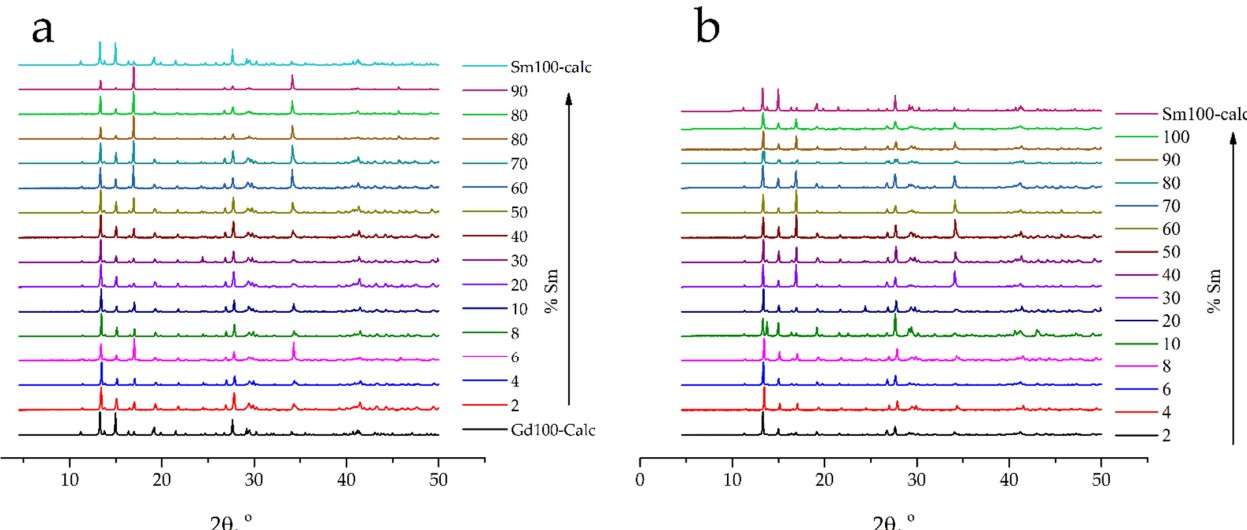

**Figure 3.** (**a**)—PXRD patterns of **Sm$_x$Gd$_{1-x}$-TDA** and simulated from single-crystal data of **Sm-TDA** and **Gd-TDA**; (**b**)—PXRD patterns of **Sm$_x$Tb$_{1-x}$-TDA** and simulated from single-crystal data of **Sm-TDA**.

The IR spectra of compounds **Sm$_x$Gd$_{1-x}$TDA** and **Sm$_x$Tb$_{1-x}$TDA** consist of bands corresponding to various stretching and deformation vibrations (Figures S1 and S2). The broadened band of stretching vibrations of OH bonds (3650–3200 cm$^{-1}$) indicates the involvement of lattice H$_2$O molecules in the network of hydrogen bonds. Symmetric vibrations ($\nu_s$) of the CH$_3$ groups occur at 3041, 2942, 2868 cm$^{-1}$. Broad stretching vibrations of the N-H bonds of the dimethylammonium cation corresponded to 2720 cm$^{-1}$. The HCOO group is characterized by stretching vibrations ($\nu_s$) at 2826 cm$^{-1}$. $\nu_s$ and asymmetric ($\nu_{as}$) vibrations of the COO$^-$ anion appear at 1560 cm$^{-1}$ and 1420 cm$^{-1}$; the difference between these frequencies indicates the manifestation of the carboxyl group of both chelate and bridging functions. Stretching vibrations of the aromatic C = C, N = N, C = N bonds of triazole ring occur at 1445, 1460, 1478, and 820 cm$^{-1}$.

### 2.2. Crystal Structure

Since all complexes are isostructural, the description is given by the example of complex **Sm-TDA**.

Structure **Sm-TDA** is based on an anionic framework [Sm(TDA)(HCOO)]$^-$ and contains NMe$_2$H$_2^+$ cations and water molecules in the cavities (Figure 4). Samarium cations are situated in a low-symmetric coordination environment. The coordination polyhedron of the central atom can be described as a distorted two-lobed trigonal prism with bases N(1), O(1), O(5); N(3), O(2), O(6) and vertices O(4), O(3). Some parameters of the crystal structures, as well as the structure of the europium complex, which was described earlier [23], are given in Table 1. It can be noted that for samarium, the unit cell parameters and bond lengths are greater, which is consistent with the radius change caused by lanthanide contraction.

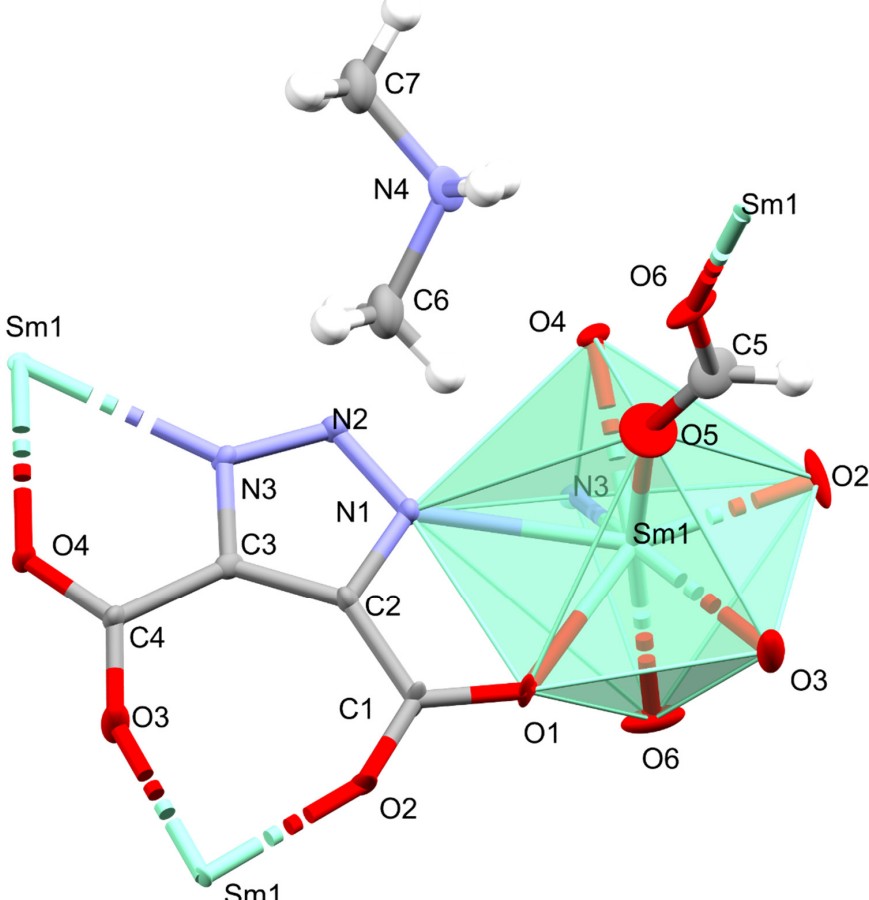

**Figure 4.** Fragment of the polymeric structure of **Sm-TDA**. The NMe$_2$H$_2^+$ cation is shown, as well as the [Sm(TDA)(HCOO)$^-$] anion. Distorted H$_2$O molecule is omitted for clarity.

**Table 1.** Comparison of some parameters of Sm, Eu, Gd complexes.

| {[NMe$_2$H$_2$][Ln(TDA)(HCOO)]} | Gd | Sm | Eu [23] |
|---|---|---|---|
| Space group | | Pna2$_1$ | |
| a, Å | 12.8145(9) | 12.8521(5) | 12.7834(4) |
| b, Å | 9.9105(6) | 10.0035(5) | 10.0323(3) |
| c, Å | 8.8666(6) | 8.9281(7) | 8.8998(4) |
| α, β, γ | | 90° | |
| 1O-Ln, Å | 2.44(1) | 2.443(7) | 2.440(7) |
| 1N-Ln, Å | 2.58(5) | 2.58(1) | 2.572(6) |
| 5O-Ln, Å | 2.36(1) | 2.36(1) | 2.31(1) |
| 6O-Ln, Å | 2.32(2) | 2.334(9) | 2.333(9) |
| 3O-Ln, Å | 2.379(6) | 2.37(4) | 2.374(4) |
| 4O-Ln, Å | 2.438(8) | 2.435(4) | 2.432(5) |
| 2O-Ln, Å | 2.33(1) | 2.33(1) | 2.36(1) |
| 3N-Ln, Å | 2.54(1) | 2.555(5) | 2.554(6) |

All TDA$^{3-}$ anions are equivalent, and each coordinates the samarium cation with two donor atoms: either two oxygen atoms (O(3) and O(2)), or an oxygen atom and a nitrogen atom (O(1), N(1) and O(4), N(3)). Formate anions, binding two samarium cations, play the role of bridging ligands. The structure of the synthesized complexes contains a system of channels, the maximum size of which is ~7 × 7 Å, which is associated with the tendency of the H$_3$TDA ligand to form framed polymer compounds. The view along the b axis of the polymer structure of the MOF is shown in Figure 5. Dimethylammonium cations in the structure are disordered, and the displacement of nitrogen atoms between the two positions is 1.255 Å. The disordering can be explained by the fact that the size of the cavities in the frame is much larger than the size of the dimethylammonium cation.

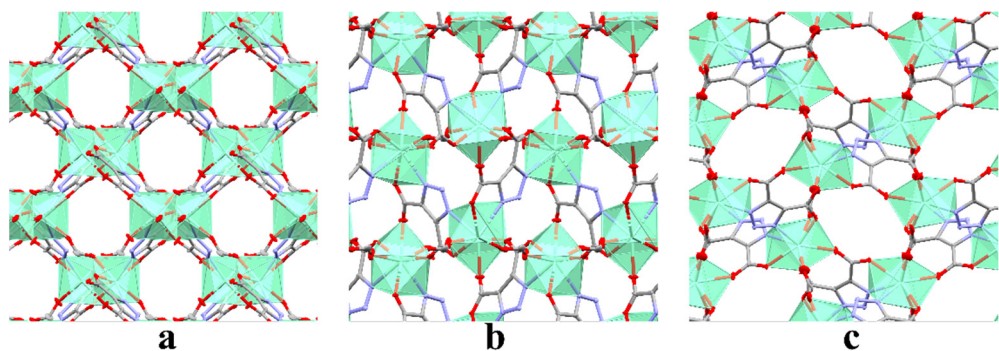

**Figure 5.** Crystal structure of **Sm-TDA**. View along the crystallographic axis (**a–c**). Hydrogen atoms and NMe$_2$H$_2$$^+$ cations are omitted for clarity.

### 2.3. Luminescent Properties

All of the obtained compounds exhibit photoluminescence upon excitation with UV light. The **Sm-Gd** compounds under UV irradiation exhibit luminescence, the color of which varies from orange-red (Sm100%), characteristic of samarium complexes, to almost white, with a large dilution of Sm$^{3+}$ ions with Gd$^{3+}$ ions. The luminescence spectra of **Sm$_x$Gd$_{1-x}$-TDA** series samples contain bands of fluorescence and phosphorescence of the ligand, with maxima at ~430 and ~550 nm (Figure 6). The fluorescent nature of the first band is confirmed by comparing the emission spectra recorded in the fluorescence and phosphorescence mode (Figure 7). A complicated oscillatory structure is observed in the fluorescence and phosphorescence bands, which indicates the high rigidity of the framework. The contribution of the fluorescent band remains significant for any Sm$^{3+}$:Gd$^{3+}$ ratios, while the contribution of the phosphorescent band is significant only at high concentrations of gadolinium. This is due to the role of Gd$^{3+}$ ions in the "paramagnetic effect", which enhances the phosphorescence of and reduces the intensity of ligand fluorescence

by increasing the rate of intersystem crossing (ISC) between the singlet and triple spin states [50–52]. This explains the unusual dependence of the color coordinates (Figure 6, inset) from the $Sm^{3+}$ fraction. Color coordinates of the sample **$Sm_{02}Gd_{98}$-TDA** (0.290, 0.356) were found to be quite close to the white color (0.333, 0.333). In addition to ligand bands, the luminescence spectrum contains bands corresponding to the samarium transitions $^4G_{5/2} \rightarrow ^4H_j$, j = 5/2 ($\lambda_{max}$ = 562 nm), 7/2 ($\lambda_{max}$ = 600 nm, the most intense), 9/2 ($\lambda_{max}$ = 643 nm), 11/2 ($\lambda_{max}$ = 705–705 nm, the least intense). The presence of both ligand and lanthanide emission bands is associated with the insufficiently effective transfer of excitation energy from the ligand to the $Sm^{3+}$ ion. It can be explained by high energy gap between the triplet level of the ligand (25,300 $cm^{-1}$ [23]) and the resonant level of the $Sm^{3+}$ ion (17,800 $cm^{-1}$ [17]), although, in the case of the europium complex studied earlier (the energy of the resonance level of 17,240 $cm^{-1}$), phosphorescence of the ligand was not observed [23]. Despite this fact, the luminescence of samarium complexes is usually less intense than that of the corresponding europium derivatives [17,32].

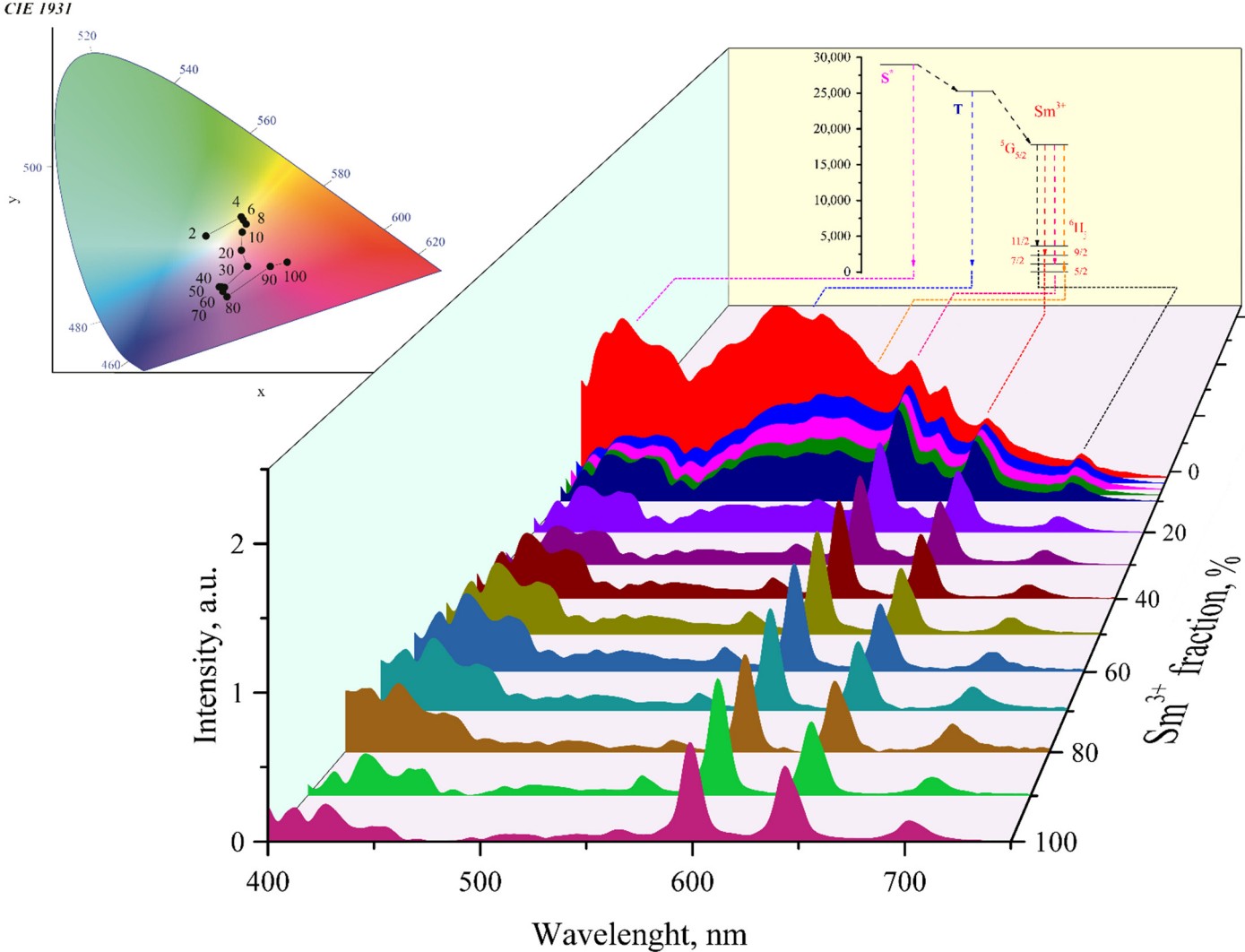

**Figure 6.** Emission spectra of $Sm_xGd_{1-x}$-TDA complexes and Jabłoński diagram. Inset: CIE color coordinates.

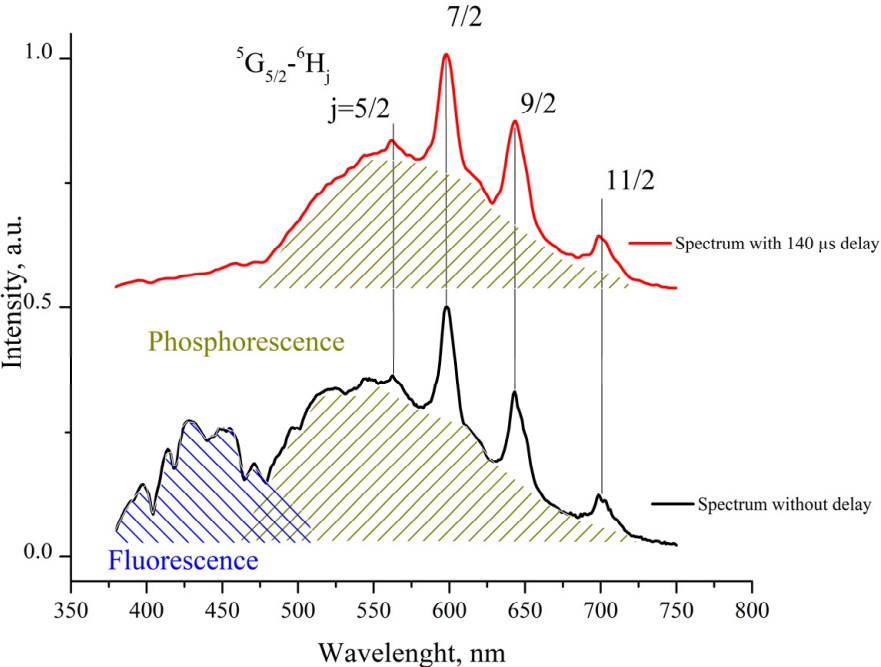

**Figure 7.** Luminescence spectra of $\mathbf{Sm_{04}Gd_{96}}$-**TDA** ($\lambda_{ex}$ 280 nm) recorded in the fluorescence (black line) and phosphorescence (red line) modes.

The kinetics of the luminescence decay of $Sm^{3+}$ in $\mathbf{Sm_xGd_{1-x}TDA}$ is satisfactorily described by the exponential law

$$I = y_0 + A_1 e^{\frac{-t}{\tau_1}} + A_2 e^{\frac{-t}{\tau_2}} \tag{1}$$

where $\tau_1$ and $\tau_2$ are the observed lifetimes of the excited state (Figure S4, Table S2). The dependence of $\tau_1$ and $\tau_2$ on the concentration of $Gd^{3+}$ ions is shown in Figure 8. In all cases, $\tau_2$ is significantly greater than $\tau_1$ in the order of magnitude of tens of μs for $\tau_2$, units of μs for $\tau_1$, and it can be concluded that $\tau_1$ refers to the luminescence of $Sm^{3+}$ ions, and $\tau_2$ is associated with the phosphorescence of the ligand. For the $Gd^{3+}$ complex, we observed the lifetime of the excited state of the order of 100 μs [23]; the drop is due to the transfer of excitation energy to the $Sm^{3+}$ ion. The value of $\tau_1$ increases almost linearly with an increase in the concentration of $Gd^{3+}$ up to a content of 60–70%, after which it drops sharply. In this case, two processes act in opposite directions: on the one hand, dilution of $Sm^{3+}$ ions with $Gd^{3+}$ ions leads to a decrease in concentration quenching; on the other hand, an increase in the proportion of $Gd^{3+}$ increases the probability of phosphorescence of the ligand, which is confirmed by luminescence spectra. The ratio of the pre-exponential factors $A_1$: $(A_1 + A_2)$ also demonstrates a dramatic transition: at the concentration of $Gd^{3+}$ more than 70%, the contribution of the ligand's own luminescence begins to dominate the $Sm^{3+}$ radiation (Figure 9).

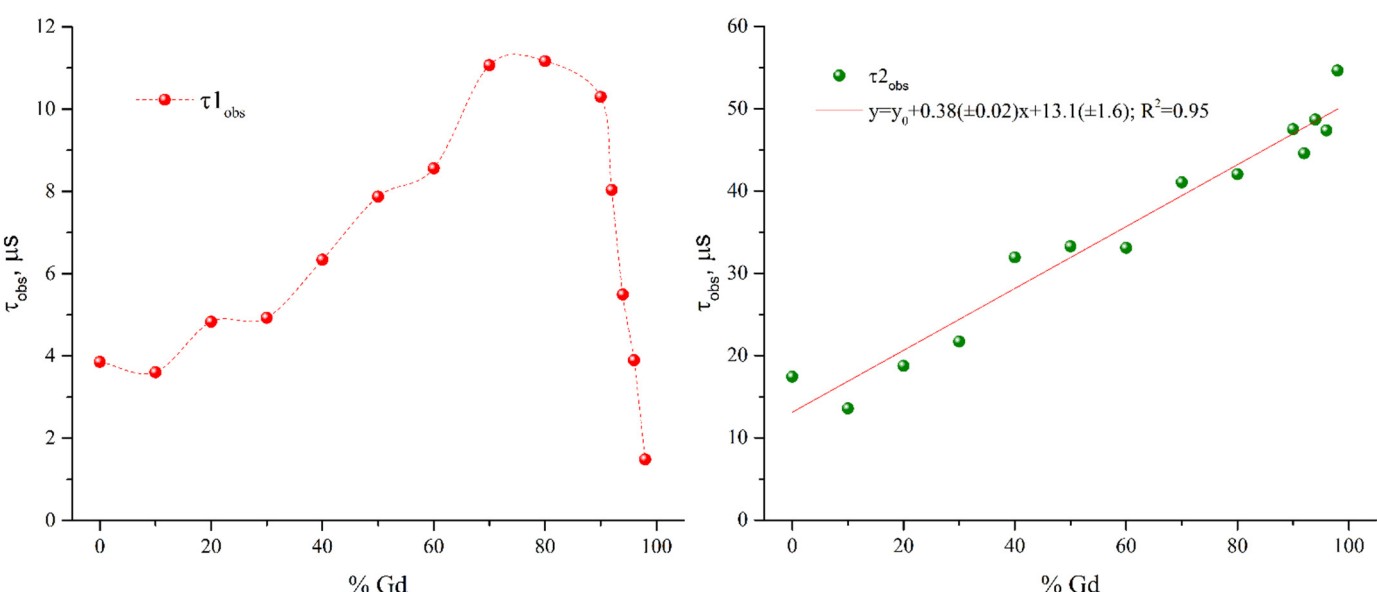

**Figure 8.** Dependence of the observed lifetime of $Sm^{3+}$ on the concentration of $Gd^{3+}$. The component $\tau_1$ on the left, the component $\tau_2$ on the right.

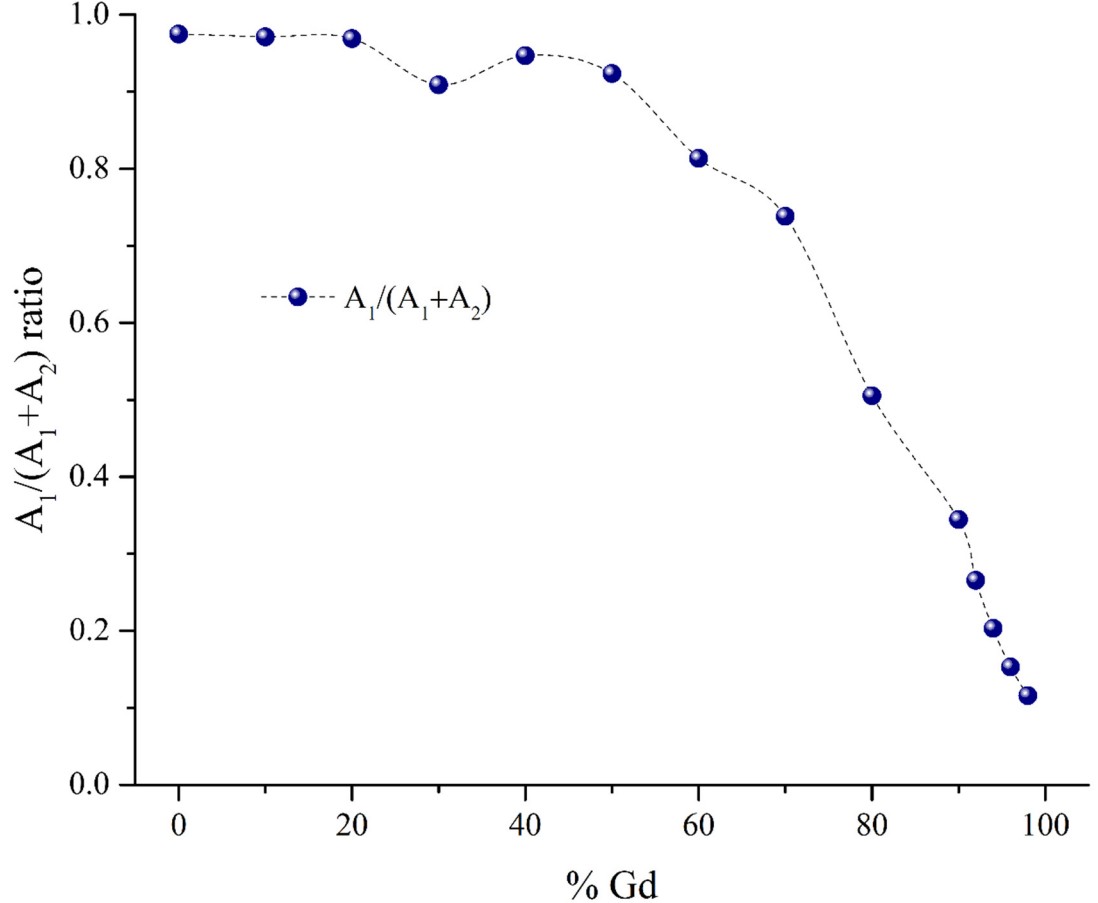

**Figure 9.** Change in the contribution of $\tau_1$ component from $Gd^{3+}$ concentration in series $Sm_xGd_{1-x}TDA$.

This phenomenon is typical for weakly emitting $Sm^{3+}$ and $Dy^{3+}$ ions [20]. Despite this, the lifetime of the excited state of the $Sm^{3+}$ ion increases with an increase in the fraction of $Gd^{3+}$. This indicates the suppression of concentration quenching.

In the luminescence spectra of the Sm-Tb compounds (**$Sm_xTb_{1-x}TDA$**) (Figure 10), the emission of both $Sm^{3+}$ and $Tb^{3+}$ ions is evident, and even with a content of $Tb^{3+}$ of 98%, the intensity of the samarium bands is comparable to those of terbium (Figure 11).

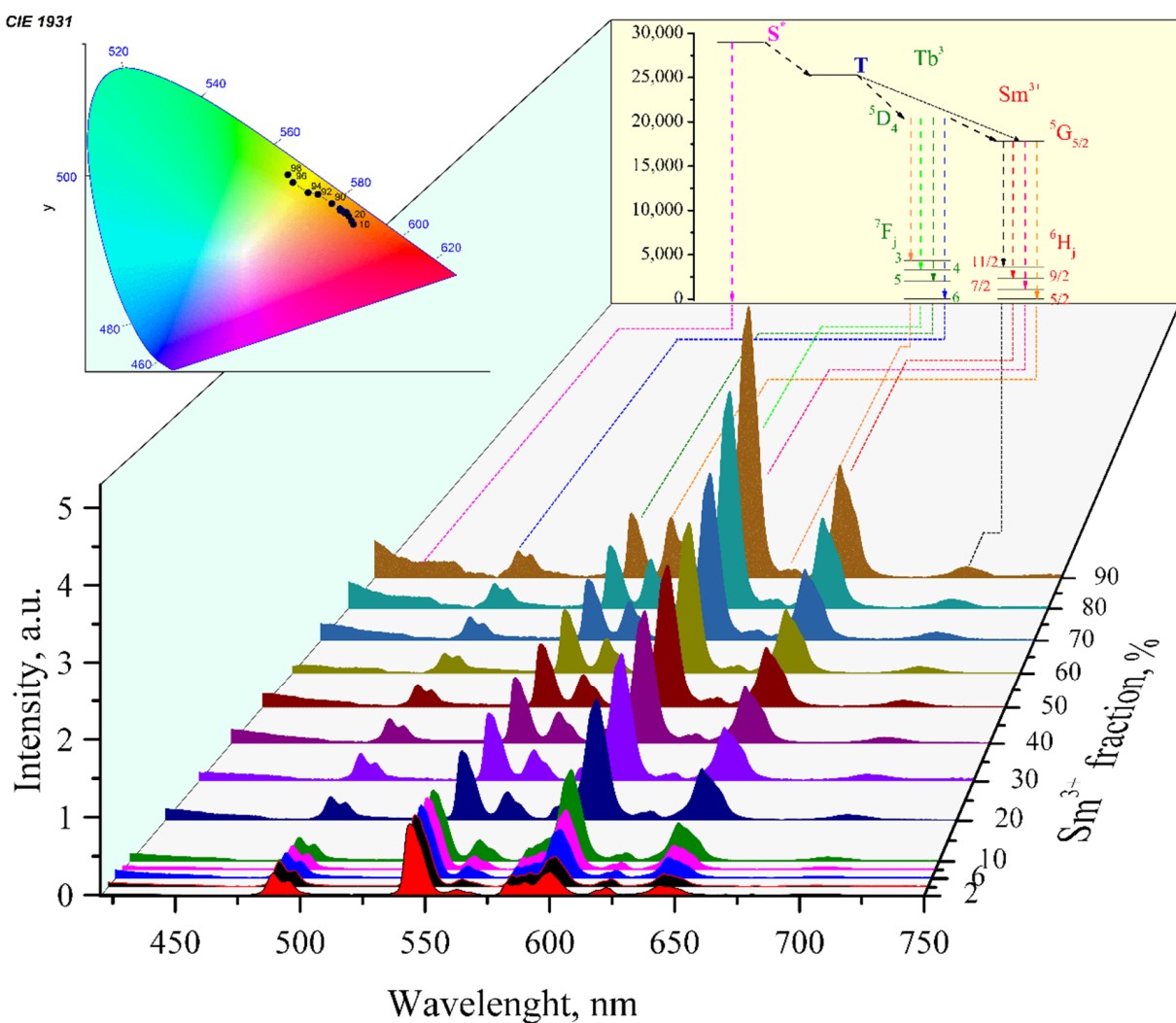

**Figure 10.** Emission spectra of **$Sm_xTb_{1-x}$-TDA** complexes and Jabłoński diagram. Inset: CIE color coordinates.

Unlike series **$Sm_xGd_{1-x}$-TDA**, the emission of the ligand is quenched, and its intensity decreases rapidly with an increase in the relative concentration of $Tb^{3+}$. This is due to the efficient transfer of energy from the triplet level of the ligand to the resonance level of $Tb^{3+}$ ions. Alongside bands corresponding to $Sm^{3+}$ transitions, the luminescence excitation spectra of $Sm^{3+}$ also contain bands corresponding to $Tb^{3+}$ transitions, which indicates the sensitization of the luminescence of samarium ions by terbium ions. Kinetic measurements confirm this hypothesis. The decay curves of $Tb^{3+}$ ions (Figure S5, Table S3) are described as biexponential dependences, both components of which increase significantly only with an increase in the fraction of terbium to 90% and higher (Figure 12).

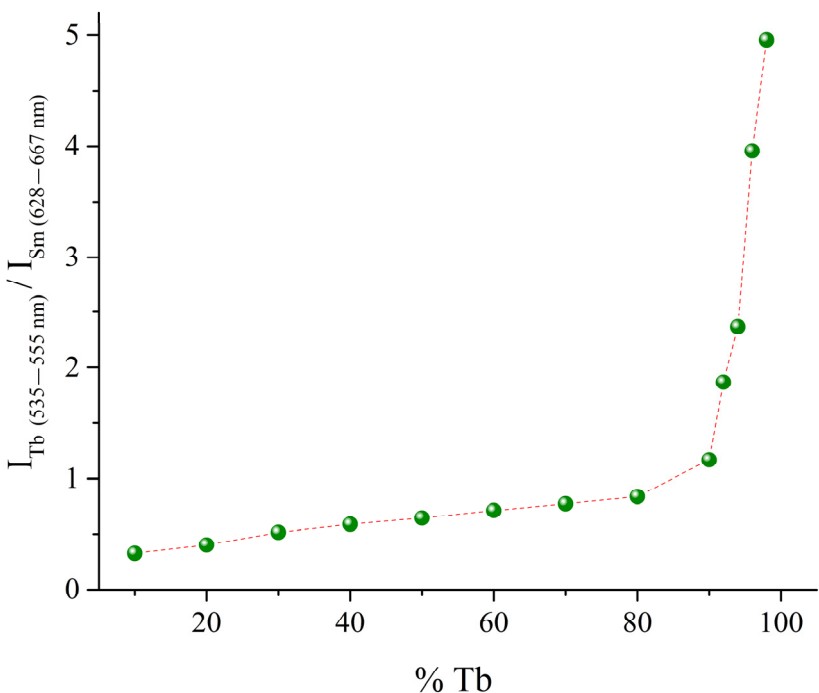

**Figure 11.** Dependence of the ratio of integral intensities of the strongest transitions of samarium ($^5G_{5/2}$-$^6H_{7/2}$) and terbium ($^5D_4$-$^7F_5$) on the fraction of terbium in the complexes **$Sm_xTb_{1-x}$-TDA**.

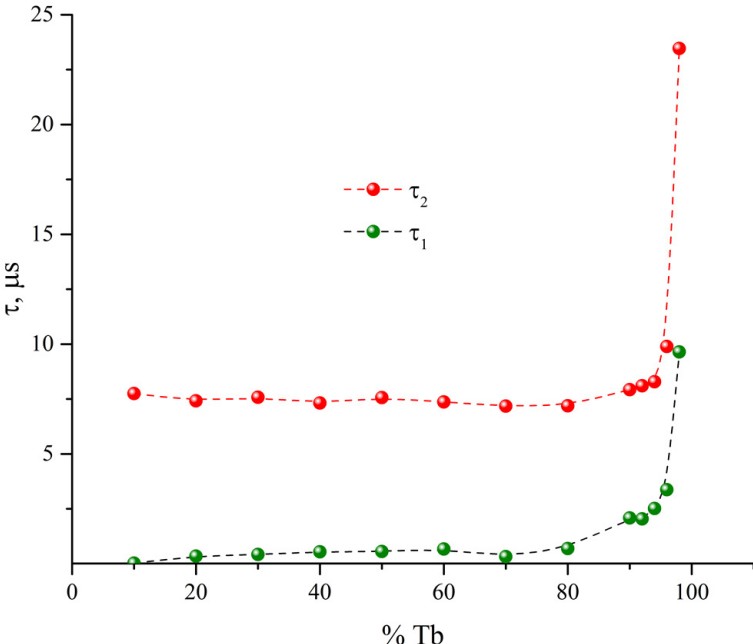

**Figure 12.** Dependence of the observed lifetime of $Tb^{3+}$ on the concentration for **$Sm_xTb_{1-x}$-TDA** series.

On the other hand, for samarium luminescence, there is an initial growth period, which we previously observed in terbium–europium bimetallic complexes [49]. The presence of this growth period makes it difficult to determine the radiative lifetime of $Sm^{3+}$ ions, but confirms the presence of the sensitization of the luminescence of samarium ions by $Tb^{3+}$ ions. For the sample with the highest Tb:Sm ratio, **$Sm_{02}Tb_{98}$-TDA**, it is possible to

successfully apply the energy transfer model proposed earlier (Figure 13). In this case, the decay of samarium luminescence is described with an equation:

$$I = y_0 + A_1 e^{\frac{-t}{\tau_1}} - A_2 e^{\frac{-t}{\tau_2}} \tag{2}$$

where the second exponent corresponds to the transfer of energy from terbium to samarium.

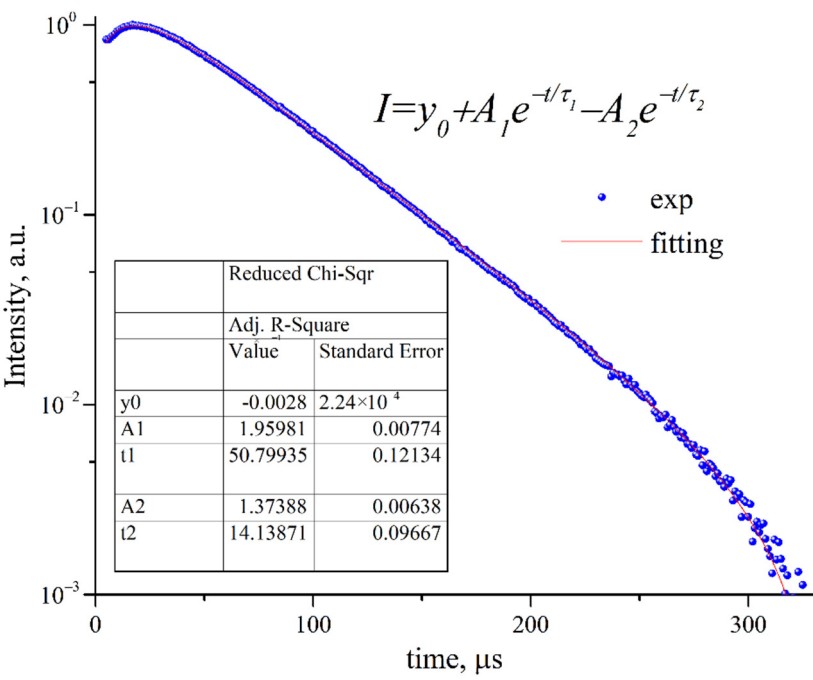

**Figure 13.** $Sm^{3+}$ luminescence decay curve for **$Sm_{02}Tb_{98}$-TDA**, $\lambda_{ex}$ 280 nm, $\lambda_{em}$ 648 nm.

This makes it possible to estimate the radiative time of samarium ($\tau_1$) as 50.8 µs, which is 34 times greater than the radiative time in the **$Sm_{02}Gd_{98}$-TDA** sample. For samples with a higher content of samarium, such an analysis cannot be carried out, since the decay process is too fast, but Figure S8 allows us to definitively state that dilution with terbium ions dramatically increases the radiative lifetime of $Sm^{3+}$ ions.

The excitation spectra of complexes **$Sm_xGd_{1-x}$-TDA** and **$Sm_xTb_{1-x}$-TDA** (Figures S4 and S5, respectively) contain both a wide band of $\pi$-$\pi^*$ transitions of the ligand (260–290 nm) and bands corresponding to direct excitation through *f-f* transitions of lanthanides. The relative intensity of the ligand excitation band is the highest at the lowest fraction of samarium (2%), while, for samples with a high metal content, the excitation through the transitions of the latter has comparable intensity. This is due to the fact that, for samples with low content of $Sm^{3+}$, the concentration of the chromophore ligand is higher, which makes the sensitization of samarium luminescence more efficient. For **$Sm_xTb_{1-x}$-TDA** compounds, when registering the luminescence at 648 nm, which corresponds to the $^5G_{5/2}$-$^6H_{9/2}$ transition of samarium, excitation is observed both through the $Sm^{3+}$ and $Tb^{3+}$ transitions. This indicates the sensitization of the samarium luminescence by terbium ions.

### 3. Materials and Methods

Commercially available reagents and solvents (Sigma-Aldrich, Burlington, MA, USA) were used. Lanthanide nitrates were obtained by the dissolving of the corresponding oxides (99.999%, LANHIT, Moscow, Russia) in concentrated nitric acid (reagent-grade, XPC). The concentration of the $Ln(NO_3)_3$ solutions was determined by complexometric titration (indicator—Xylenole Orange). Solutions were prepared using deionized (18 MΩ·cm) water.

IR spectra were recorded on an IR-Fourier spectrometer FT-IR Spectrum One Perkin-Elmer spectrum in KBr tablets of the 400–4000 $cm^{-1}$ region with a resolution of 0.5 $cm^{-1}$.

Thermal analysis was carried out on a NETZSCHSTA 409 PC/PG device in an air atmosphere, at a heating rate of 10 °C/min. Powder diffractograms were recorded using a diffractometer of the Bruker D8 Advance model (CuK$\alpha$, $\lambda$ = 1.5418 Å).

ICP-MS was performed using an inductively coupled plasma mass spectrometer ELAN mod. 9000, DRC II, DRC-e. To carry out the analysis, a sample of ~0.2000 g of the complex was calcinated in air in a porcelain crucible at 600 °C; the remainder was quantitatively transferred to 10 mL volumetric flasks and dissolved in 3–5 mL of distilled $HNO_3$, after which the solutions were diluted the required number of times. Calibration solutions were prepared based on $Ln(NO_3)_3*6H_2O$ crystallohydrates, setting the concentration by complexometric titration. To reduce the errors associated with the instability of the solution inside the ionizer, an internal standard with a solution of rhodium sulfate with a concentration of $1.13 \cdot 10^{-2}$ M was used.

The pH of the solutions was established using the pH meter I-160MI (Measuring Equipment, Moscow, Russia).

Photoluminescence spectra and lifetimes of excited states of powdered samples were measured using a Horiba-Jobin-Yvon Fluorolog-QM spectrofluorimeter (Palaiseau, France) equipped with a 75 W ArcTune xenon lamp and a Hamamatsu R-FL-QM-R13456 photomultiplier sensitive in the radiation range of 200–980 nm.

The data were collected using a Bruker D8 Quest diffractometer with a Photon III detector at a temperature of 100 (2) (**Sm-TDA**) and 120(2) (**Gd-TDA**) K, MoK$_\alpha$ radiation ($\lambda$ = 0.71073 Å), $\varphi$ and $\omega$ scanning mode. Single crystals of compounds **Sm-TDA** and **Gd-TDA** were obtained as a result of solvothermal synthesis. The structure was solved with direct methods and refined by the least-squares method in the full-matrix anisotropic approximation on $F^2$ with the use of SHELXTL и Olex2 software packages [53–55]. Crystals of **Sm-TDA** and **Gd-TDA** are racemic twins with a component ratio of 0.55:0.45 (**Sm-TDA**) and 0.60:0.40 (**Gd-TDA**). In addition, the crystal structures are pseudosymmetric due to the arrangement of heavy samarium and gadolinium atoms, approximately (by 87%, according to checkCif) corresponding to the more highly symmetric Pnma group. The hydrogen atoms were placed in the calculated positions and refined using the "riding" model. The crystallographic data, experimental parameters, and structural refinements are given in Table S1. The atomic coordinates, bond lengths, valence angles, and thermal displacement parameters are deposited in the Cambridge Structural Data Bank (CCDC No. 2097364, No. 2177609).

***The synthesis of H$_3$TDA*** was carried out according to the modified method described for 1-methyl-[1*H*]-1,2,3-triazole-4,5-dicarboxylic acid [56]:

First, 5.96 g (50.0 mmol) of [1*H*]-1,2,3-benzotriazole was dissolved in 350 mL of distilled water. After heating the solution to 80 °C with constant stirring, 44.3 g (282 mmol) of potassium permanganate was added in small portions. A day later, at reduced pressure, the solution was separated from the precipitated $MnO_2$ by filtration. The filtrate was acidified with concentrated hydrochloric acid to pH = 1. The white crystalline precipitate of triazole dicarboxylic acid was filtered at reduced pressure and dried. The yield was 70%. The melting point was 200 °C (with decomposition), which coincides with the literature data [57]. IR (KBr, $\nu$, cm$^{-1}$): 3542 s, 2900–2400 m, 1720 m, 1585 m, 1534, 1395 w, 1295 m, 995 s; NMR $^{13}$C (DMSO-d$_6$; $\delta$, m.d.): 161.5, 137.8.

***Synthesis of complexes*** **Sm-TDA**, **Gd-TDA**, **Sm$_x$Ln$_{1-x}$-TDA** x = 0.1 (Sm$_{0.9}$Ln$_{0.1}$-TDA), x = 0.2 (Sm$_{0.8}$-Ln$_{0.2}$-TDA) ... Sm$_{0.1}$Ln$_{0.9}$-TDA, Ln=Tb, Gd was carried out according to a modified method of synthesis of the europium complex [23]: in a Teflon container (10 mL) was placed 78.5 mg (0.5 mmol) H$_3$TDA, 3.00 mL DMF, and 2.5 mL 0.2 M Sm(NO$_3$)$_3$ solution or required volumes of Sm(NO$_3$)$_3$ (0.2 M) and Gd(NO$_3$)$_3$ (0.2 M) or Tb(NO$_3$)$_3$ (0.2 M) solutions. A 5% HNO$_3$ solution was added until pH 1.5 was reached. The container was sealed, placed in a steel autoclave, heated to 160 °C (50 °C/h), and kept at 160 °C for 48 h. Then, the container was cooled (2 °C/h) to room temperature. The resulting white precipitate was filtered under vacuum, washed with water (5 × 10 mL) and ethanol (2 × 10 mL), and dried in vacuo over P$_4$O$_{10}$ for 48 h.

## 4. Conclusions

In summary, as a result of replacing water with a water–DMF mixture during solvothermal synthesis, new MOFs based on [1*H*]-1,2,3-triazole-4,5-dicarboxic acid containing dimethylammonium cations and formate anions were obtained. The resulting compounds do not contain water or DMF in the inner coordination sphere of the REE, which prevents the vibrational quenching of the samarium complex. Mixed-metal systems **Sm$_x$Gd$_{1-x}$-TDA** and **Sm$_x$Tb$_{1-x}$-TDA** demonstrate multicomponent emission, which includes, in addition to the emission of samarium, in the first case, the fluorescence and phosphorescence of the ligand, and in the second case, the emission of terbium. This allows one to finely tune the color coordinates of the luminescence by changing the ratio of lanthanides. In addition to the color coordinates, there is also a change in the lifetime of the excited state of samarium, which increases with an increase in the concentration of gadolinium and terbium ions, which is associated with both the suppression of concentration quenching and the sensitization of samarium luminescence by terbium ions.

**Supplementary Materials:** The following supporting information can be downloaded at: https: //www.mdpi.com/article/10.3390/inorganics10080104/s1, Table S1 Selected crystal data and parameters for structure refinement of the **Sm-TDA** and **Gd-TDA**; Table S2 Sm$^{3+}$ luminescence fitting parameters for **Sm$_x$Gd$_{1-x}$-TDA**; Table S3 Tb$^{3+}$ luminescence fitting parameters for **Sm$_x$Tb$_{1-x}$-TDA**; Figure S1 IR spectra of complexes **Sm$_x$Gd$_{1-x}$-TDA** and free H$_3$TDA ligand; Figure S2 IR spectra of complexes Sm$_x$Tb$_{1-x}$-TDA and free H ligand; Figure S3 Excitation spectra of **Sm$_x$Gd$_{1-x}$-TDA** complexes (λem 648 nm); Figure S4 Sm$^{3+}$ decay curves for **Sm$_x$Gd$_{1-x}$-TDA**; λex = 280 nm, λem = 648 nm; Figure S5 Tb$^{3+}$ decay curves for **Sm$_x$Tb$_{1-x}$-TDA**; λex = 280 nm, λem = 545 nm; Figure S6 Sm$^{3+}$ decay curves for **Sm$_x$Tb$_{1-x}$-TDA**; λex = 280 nm, λem = 648 nm.

**Author Contributions:** Conceptualization, Y.A.B., A.A.I. and A.M.L.; formal analysis, A.A.I., I.A.A. and A.V.S.; investigation, A.A.I., A.V.S., V.E.G. and I.A.A.; resources, Y.A.B. and I.V.T.; data curation, Y.A.B., V.E.G. and A.M.L.; writing—original draft preparation, A.A.I. and Y.A.B.; writing—review and editing, Y.A.B., V.E.G., A.A.I., A.M.L. and I.V.T.; visualization, Y.A.B., A.M.L. and A.V.S.; supervision, Y.A.B. and I.V.T.; project administration, Y.A.B.; funding acquisition, Y.A.B. and I.V.T. All authors have read and agreed to the published version of the manuscript.

**Funding:** The reported study was funded by RFBR, project number 19-03-00263. Photophysical measurements were supported by the Russian Science Foundation, project number 19-13-00272.

**Institutional Review Board Statement:** Not applicable.

**Informed Consent Statement:** Not applicable.

**Data Availability Statement:** Not applicable.

**Acknowledgments:** This work was supported in part by the M.V. Lomonosov Moscow State University Program of Development. The authors thank Anna M. Shmykova for her help in preparing the graphical abstract.

**Conflicts of Interest:** The authors declare no conflict of interest.

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
