# Peer review of "New Carboxylate Anionic Sm-MOF: Synthesis, Structure and Effect of the Isomorphic Substitution of Sm3+ with Gd3+ and Tb3+ Ions on the Luminescent Properties"

_inorganics, doi:10.3390/inorganics10080104_

Round 1

Reviewer 1 Report

The manuscript submitted by Ivanova presents a very nice piece of work related to a topical filed of lanthanide-based luminophores. The research is well designed and obviously deserves publication. However, some critical notes should be considered by authors team before acceptance: 

1. Cif file for Sm compound must be resubmitted to CCDC after  structure re-refinement to be converged and filling missing crystal size, color and habit fields. 

2. ESI file needs to be added to submission. 

3. x and 1-x indices are confused in line 18

4.  intra-atmospheric is possibly needed in line 56

5. 2.1 → 2.2 in line 146

6. Some empty space is recommended to introduce between the figures in line 162

7. Spacebars between values and their deviations should be deleted throughout table 1. 

8. Please check lines 354-356 as the sense of this part seems a bit elusive. 

Author Response

The manuscript submitted by Ivanova presents a very nice piece of work related to a topical filed of lanthanide-based luminophores. The research is well designed and obviously deserves publication.

Thank you for your review, comments and good evaluation of our article!

However, some critical notes should be considered by authors team before acceptance: 

  1. Cif file for Sm compound must be resubmitted to CCDC after  structure re-refinement to be converged and filling missing crystal size, color and habit fields.

Thanks for the comments! All necessary changes have been made;  corrected file was  re-deposited.

  1. ESI file needs to be added to submission. 

Thank you for your comment! The ESI file is attached.

  1. x and 1-x indices are confused in line 18
  2. intra-atmospheric is possibly needed in line 56
  3. 2.1 → 2.2 in line 146
  4. Some empty space is recommended to introduce between the figures in line 162
  5. Spacebars between values and their deviations should be deleted throughout table 1. 

Thank you for your comments!  All these remarks have been corrected in the text of the work.

  1. Please check lines 354-356 as the sense of this part seems a bit elusive. 

Thank you for your comment! The error has been fixed.

in a teflon container (10 ml) was placed 78.5 mg (0.5 mmol) H3TDA, 3.00 ml DMF and 2.5 ml 0.2 M Sm(NO3)3 solution or required volumes of Sm(NO3)(0.2 M) and Gd(NO3)3 (0.2 M) or Tb(NO3)(0.2 M) solutions.

Reviewer 2 Report

This manuscript reported the synthesis, structure, and the luminescent properties of two new carboxylate anionic Sm-MOFs. In particular, the influence of isomorphic substitution of Sm3+with Gd3+and Tb3+ionson its luminescent properties are explained. When Gd3+ and Tb3+ ions with different optical activities are used for substitution, the spectra and decay kinetics of Sm-TDA and Gd-TDA are different, which is interesting. The experiment and characterization are comprehensive; however, many important problems need to be revised urgently and the manuscript is recommended for publication after some revisions as follows:

1.      "Isomorphyc" in the title is misspelled and should be "isomorphic".

2.      PXRD data (Fig. S1 and S2) can be considered in the main body of this paper as important evidence for proving the synthesis of SmxGd1-x-TDA andSmxTb1-x-TDA pure phases. Please further explain the mechanism of pH on the formation of Sm-TDA single crystal.

3.      There is a minor error, label “2.1 Crystal structure” should be "2.2 …".

4.      It is very interesting that the results of the contribution of the phosphorescent band is significant only at high concentrations of gadolinium. But unfortunately, the author only understands it as the "paramagnetic effect" of Gd3+ ions on TDA3- ligand, which lacks theoretical depth. Please further explain this phenomenon in combination with previous literature.

5.      Compared with Gd3+ doping, the emission spectra of SmxTb1-x-TDAare completely different, especially the luminescence behavior of ligands. Please explain in detail the reason why Tb3+ doping leads to ligand luminescence quenching. Is it related to the optical activity of doped ions?

6.      Due to the optical activity of Tb3+ and the sensitization of Tb3+ to Sm3+, it shows complex decay kinetics. Moreover, in the case of low Tb3+ concentration, there is obviously phosphorescence of ligand in the spectrum. When using double exponential function fitting, it is difficult to explain τ1 and τ2clearly in physics. Please reviewed the rationality of using two-component fitting.

Author Response

This manuscript reported the synthesis, structure, and the luminescent properties of two new carboxylate anionic Sm-MOFs. In particular, the influence of isomorphic substitution of Sm3+with Gd3+and Tb3+ionson its luminescent properties are explained. When Gd3+ and Tb3+ ions with different optical activities are used for substitution, the spectra and decay kinetics of Sm-TDA and Gd-TDA are different, which is interesting. The experiment and characterization are comprehensive; however, many important problems need to be revised urgently and the manuscript is recommended for publication after some revisions as follows:

Thank you for your review, comments and good evaluation of our article!

  1. "Isomorphyc" in the title is misspelled and should be "isomorphic".

Thank you for your comment! The error has been fixed.

  1. PXRD data (Fig. S1 and S2) can be considered in the main body of this paper as important evidence for proving the synthesis of SmxGd1-x-TDA andSmxTb1-x-TDA pure phases.

Thank you for your comment.

According to your recommendation, figures S1 and S2 have been moved to the article (Figure 3 a and b).

 Please further explain the mechanism of pH on the formation of Sm-TDA single crystal.

Thanks for the interesting question! According to our assumption, an increase in acidity leads to a decrease in the rate of formation of MOF crystals, which improves their quality. The corresponding explanation has been added to the text

An decrease of the pH of the solution contributes to a decrease in the concentration of the ligand anion TDA3-, which slows down the MOF formation reaction and promotes the growth of larger crystals suitable for X-ray

  1. There is a minor error, label “2.1 Crystal structure” should be "2.2 …".

Thank you for your comment! The error has been fixed.

  1. It is very interesting that the results of the contribution of the phosphorescent band is significant only at high concentrations of gadolinium. But unfortunately, the author only understands it as the "paramagnetic effect" of Gd3+ ions on TDA3- ligand, which lacks theoretical depth. Please further explain this phenomenon in combination with previous literature.

The paramagnetic atom effect, manifested in particular for paramagnetic lanthanide chelates, is associated with an increase in the rate of intersystem crossing (ISC) due to partial mixing of singlet and triplet levels. Due to this effect, when gadolinium ions are introduced, the relative intensity of ligand fluorescence decreases, and phosphorescence increases. The corresponding explanation has been added to the text.

The contribution of the fluorescent band remains significant for any Sm3+:Gd3+ ratios, while the contribution of the phosphorescent band is significant only at high concentrations of gadolinium. This is due to the role of Gd3+ ions in the "paramagnetic effect", which enhances the phosphorescence of and reduces the intensity of ligand fluorescence by increasing the rate of intersystem crossing (ISC) between the singlet and triple spin states [50-52]

  1. Tobita, S.; Arakawa, M.; Tanaka, I. Electronic Relaxation Processes of Rare Earth Chelates of Benzoyltrifluoroacetone. J. Phys. Chem. 1984, 88, 2697–2702, doi:10.1021/j150657a006.
  2. Tobita, S.; Arakawa, M.; Tanaka, I. The Paramagnetic Metal Effect on the Ligand Localized S1 .Apprx. .Fwdarw. T1 Intersystem Crossing in the Rare-Earth-Metal Complexes with Methyl Salicylate. J. Phys. Chem. 1985, 89, 5649–5654, doi:10.1021/j100272a015.
  3. Huang, H.; Bu, Y. Effect of Paramagnetic Open-Shell Gadolinium(III) Texaphyrin on Its Kinetics and Electronic Structures in Fluorescence and Phosphorescence Emission States. J. Phys. Chem. C 2019, 123, 28327–28335, doi:10.1021/acs.jpcc.9b05453.

5.      Compared with Gd3+ doping, the emission spectra of SmxTb1-x-TDAare completely different, especially the luminescence behavior of ligands. Please explain in detail the reason why Tb3+ doping leads to ligand luminescence quenching. Is it related to the optical activity of doped ions?

      The energy of the resonant level of Gd3+ ions is approximately 32,200 cm-1, while for Tb3+ resonant energy is approximately 20,500 cm-1. Due to this, gadolinium's own luminescence in complexes is usually not manifested, and for terbium it is extremely common. As a result, instead of ligand phosphorescence in the case of terbium-samarium complexes, energy transfer from the ligand to terbium ions and from the latter to samarium ions is observed, i.e., "stepwise" sensitization of luminescence (LigandàTb3+àSm3+. Relevant explanations have been added to the text of the work.

This is due to the efficient transfer of energy from the triplet level of the ligand to the resonance level of Tb3+ ions

  1. Due to the optical activity of Tb3+ and the sensitization of Tb3+ to Sm3+, it shows complex decay kinetics. Moreover, in the case of low Tb3+ concentration, there is obviously phosphorescence of ligand in the spectrum. When using double exponential function fitting, it is difficult to explain τ1 and τ2clearly in physics. Please reviewed the rationality of using two-component fitting.

In the case of Tb3+ luminescence decay curves, registration was carried out at a wavelength of 545±5 nm corresponding to the terbium transition 5D4-7F5. In this region, neither ligand fluorescence nor Sm3+ emission not observed. In addition, the characteristic fluorescence time of ligands, as a rule, is significantly less (about 1-50 ns) than the values which can be measured on a device with a pulsed xenon lamp. Based on these assumptions, we believe that both components of the biexponential decay is related exclusively to Tb3+.

Biexponential decay is usually attributed to systems with 2 crystallographically independent luminescent centers, but this is not always obvious. In the case of our complexes, we assume that due to the presence of a large number of close metal-metal distances in the structure (about 8 "neighboring" Ln3+ ions at distances of 6-8Å), there may have 0, 1, 2 or more samarium ions among the "neighbors" of each terbium atom. Terbium cations containing samarium ions among their "neighbors" effectively sensitize the luminescence of the latter, as a result of which their τobs decreases. Those terbium cations that do not have such "neighbors" lose excitation energy less efficiently and exhibit greater τobs.

This explanation is one of many possible.  The literature describes some examples of compounds containing lanthanide cations with the only one crystallographically independent type, but exhibiting bi- or polyexponential decay (see, for example, https://doi.org/10.1021/acs.inorgchem.7b02071; https://doi.org/10.1016/j.jssc.2018.08.017 etc).

Round 2

Reviewer 2 Report

The authors have revised the manucsript acoording to the reviewer reports. The current version can be accept for publication.